# Pelvic neural injuries and acute voiding changes in rat models of radical hysterectomy

**Miaomiao Liu**[1,2☯]**, Lin Qian**[3☯]**, Haibin Wei**[3]**, Jing Zhang**[4]**, Ting Wang**[5]*****,
**Yanpeng Wang** [1,2,6]*

1 Department of Postgraduate Education, Jinzhou Medical University, Jinzhou, Liaoning Province, China, 2 Center for Reproductive Medicine, Department of Gynecology, Zhejiang Provincial People's Hospital, Affiliated People's Hospital, Hangzhou Medical College, Hangzhou, Zhejiang, China, 3 Urology & Nephrology Center, Department of Urology, Zhejiang Provincial People's Hospital, Affiliated People's Hospital, Hangzhou Medical College, Hangzhou, Zhejiang, China, 4 Center for Reproductive Medicine, Department of Obstetrics, Zhejiang Provincial People's Hospital, Affiliated People's Hospital, Hangzhou Medical College, Hangzhou, Zhejiang, China, 5 Cancer Center, Department of Pathology, Zhejiang Provincial People's Hospital, Affiliated People's Hospital, Hangzhou Medical College, Hangzhou, Zhejiang, China, 6 Center for Reproductive Medicine, The First Affiliated Hospital, Zhejiang University School of Medicine, Hangzhou, Zhejiang, China

☯ These authors contributed equally to this work.
\* 15649032616@139.com (TW); wangyanpeng@hmc.edu.cn (YW)

## Abstract

### Objective

To establish experimental models of radical hysterectomy based on Querleu-Morrow classification, and clarify the quantitative evaluation of pelvic neural injuries and acute voiding changes postoperatively.

### Methods

Female Sprague Dawley rats were randomized and received sham operation, type A, B1, C1 and C2 radical hysterectomies (as the injury gradually increased), respectively. The excised specimens were collected for hematoxylin and eosin staining and Pgp9.5 (pan-neuronal marker) immunohistochemistry to evaluate the facial and neural resection of paracervix. At 21 days after operation, 5 rats in each group were used for urine spot test, awake cystometry and leak point pressure test, and the other 5 ones were used for hematoxylin and eosin staining of bladder and pelvic neural plane, and Masson's trichrome staining of bladder.

### Results

Paracervical Pgp9.5 immunohistochemistry revealed that the resected neural area in C2 group was significantly larger than that in type A, B1, and C1 groups. Compared with type A and B1 groups, the excised paracervical facial area was significant higher in type C1 and C2 groups. The occurrence of urinary retention was 0%, 10%, 40% and 100% in type A, B1, C1 and C2 groups, respectively, which was further confirmed by average residual volume. The incidence of neurogenic bladder and its severity gradually increased from type A to type C2 groups, consistent with the findings of leakage point pressure, bladder size, bladder weight,

**Data Availability Statement:** The data are within the paper and Supporting Information files. It is also available from https://doi.org/10.6084/m9.figshare.27111208.

**Funding:** National Natural Science Foundation of China (81971363), Natural Science Foundation of Zhejiang Province (LY22H040008), and Zhejiang Traditional Chinese Medicine Administration (2021ZB021).The funders had no role in study design, data collection and analysis, decision to publish, or preparation of the manuscript.

**Competing interests:** The authors have declared that no competing interests exist.

pathological changes and collagen deposition. Neuropathological evaluation revealed neural injuries involved the main components of pelvic neural plane.

## Conclusion

The novel rat models of radical hysterectomy based on Querleu-Morrow classification revealed the structural and functional changes of voiding after operation, which reflected the situation in humans.

## 1. Introduction

Cervical cancer is the fourth most common cancer and the fourth leading cause of cancer death in women worldwide [1]. Since initially developed in 20th century, radical hysterectomy (RH) has been the mainstay of treatment for patients with early-stage cervical cancer. Based on the lateral extent of resection, the Querleu-Morrow (Q-M) classification divided RH into four major types (A-D), including a few subtypes when necessary [2–4]. Type A is an extra-fascial hysterectomy, and type B1 is the modified RH in which the transection of the paracervix is at the level of the ureteral tunnel, with no intention to excise the bladder nerves. Type C1 is a nerve-sparing radical hysterectomy (NSRH), in which the parametrium is resected at the level of the internal iliac vein, the bladder, and the rectum. Type C2 is a classical extensive hysterectomy without preservation of nerves. Type D is a laterally extended resection of the entire paracervix at the pelvic sidewall. Rather than only focusing on the length of resection, Q-M classification emphasizes the three-dimensional anatomic template for parametrial resection and definite resection landmarks [3], which makes it the most popular classification for RH worldwide [5, 6].

RH typing, including Q-M classification, was conceived mainly on cadaver studies. The human pelvic plexus is composed by fusion of pelvic splannic nerves (S2, S2, S4 nerves) and hypogastric nerve, and gives branches to uterine nerve, vesical nerve and rectal nerve innervating corresponding targeting organs [7]. The pelvic plexus in rat has similar structures to humans, with some difference of pelvic splannic nerve origin (L6 and S1) [8]. It is believed that as the surgical excision expanded, RH may damage the pelvic plexus and lead to urination, defecation and sexual disorders [9]. However, due to the lack of experimental models and some obstacles in clinical anatomy, comprehensive animal study on RH was rare, and the quantitative evaluation of pelvic neural resection and voiding behavior after RH, and the neural and vesical pathological changes post RH remain to be clarified.

An extra paracervical lymphadenectomy in Type B2 RH causes no more neural injury than type B1 RH, and laterally extended resection in type D RH destroys pelvic plexus and increases mortality [3, 4]. Thus, in this study, we tried to develop rat models of type A, B1, C1, and C2 RHs according to the Q-M classification principles. We demonstrated the structural characteristics of different types of RH, and provided distinct clues of neural injury and acute voiding dysfunction after RH. Our results may pave the way for further studies on neural injury and voiding dysfunction after RH in humans.

## 2. Materials and methods

### 2.1. Animals and ethical approval

All experimental procedures were conducted in strict accordance with the Chinese National Guidelines (GB/T 35892–20181) for the care and use of laboratory animals. Animal Ethics

Committee of Zhejiang Provincial People's Hospital approved this study. 12-week-old healthy SPF-grade female Sprague-Dawley (SD) rats (n = 63, 280–320 g) were used.

## 2.2. Experimental design

Fifty SD rats were randomly divided into 5 groups, and received sham operation, type A, B1, C1, and C2 hysterectomy respectively. All surgeries were performed under isoflurane anesthetic, with all efforts made to minimize animal suffering, and by the same experienced surgeon. Excised cervices (n = 5) were fixed with 4% paraformaldehyde and sent for hematoxylin and eosin (H&E) staining and immunohistochemistry (IHC). The residual urine volume was examined at 3, 7, 14 and 21 days after operation. During experimental periods, animals were monitored 3 times per day for potential signs of suffering, mainly weight loss of more than 20% and significant changes in animals' behavior, body posture or respiration. Rats with signs of suffering or observed for 21 days following the operation were euthanized by anesthesia with an overdose of isoflurane in order to prevent further suffering.

In C2 group, 2 rats died at 2 days after operation, 5 rats suffering from severe urinary retention were euthanized from 5 days to 14 days after operation, and only 3 rats survived to 21 days after operation. As a consequence, we performed an alternative uterus-preserving neural resection (type C2N, n = 10) for further evaluation. One rat in type B1 group and 2 rats in type C1 group died at 2 to 3 days after operation due to operation complication, and additional rats were supplemented accordingly.

At 21 days after operation, 5 rats in each group were used for urine spotting test, leak point pressure test, and awake cystometry. The other 5 rats were used for histological examination. H&E staining was performed on main components of pelvic neural plane (PNP), e.g., hypogastric nerve (HGN), pelvic splanchnic nerve (PSN), major pelvic ganglion (MPG), and vesical nerve branch (VNB). The excised bladder was weighted and then sent for H&E, Masson's trichrome, and immunohistochemical stainings.

## 2.3 Surgical procedures

All surgeries were done by the same experienced surgeon. Based on pelvic nerve anatomy in rat (Fig 1A) and Q-M classification, the corresponding rat RH for type A, B1, C1, C2 and C2N were established.

The rats were anesthetized with 1.5% isoflurane (RWD Life Science Co., Ltd, Shenzhen, China), the lower abdomen and vagina were disinfected with povidone iodine, and prophylactic cephalosporin of 5mg/kg (Hefei Dragon God Animal Pharmaceutical Co., Ltd. Hefei, China) was given intramuscularly. An approximately 3 cm long mid-line incision was created on the lower abdomen. The "Y" shaped uterus was identified and retracted contra-laterally and rostrally, the parametric adipose tissue was retracted laterally, and the bladder was retracted contra-laterally and caudally.

(1) For type A RH, the uterine horn was separated from the ovary after coagulating and cutting the oviduct and communication vessels in the mesosalpinx with a bipolar forceps. The uterine artery and vein were identified, coagulated and cut at the lower uterine segment adjacent to the uterus after excising mesometrium. The paracervix was resected halfway between the cervix and ureter, the bladder was separated from the cervix after opening vesical peritoneal reflection, and the dorsal parametrium was excised minimally. The uterus was resected from the vagina at the fornix, and vaginal cuff was closed with 6–0 prolene.

(2) For type B1 RH, a peritoneal incision was made media to the mesoureter, and the ureter was freed from parametrium. The bladder was separated from the cervix and vaginal fornix after opening vesicouterine peritoneal reflection. After that, a small initial part of the

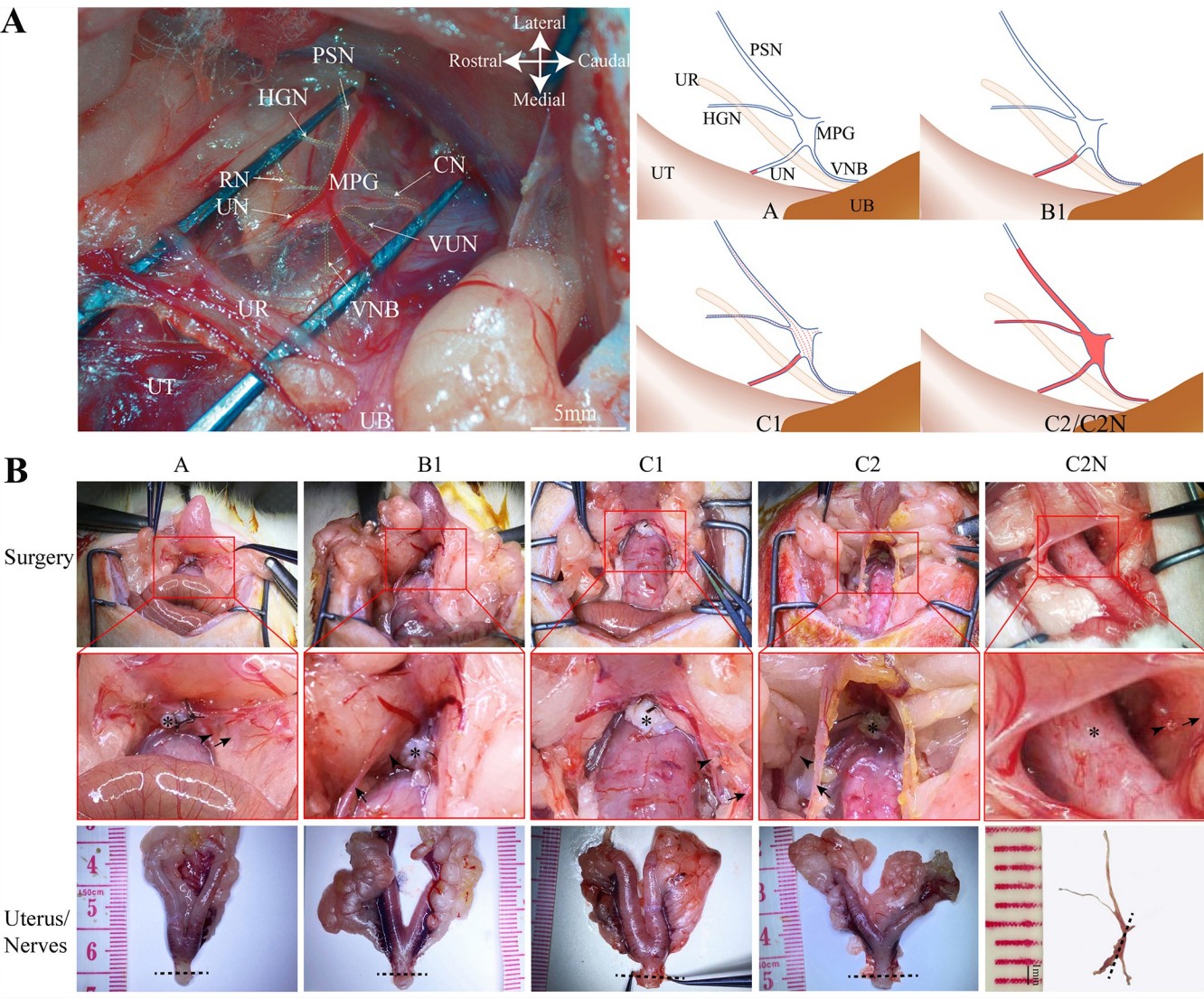

**Fig 1. Surgical procedures. A**, the main structures of pelvic neural plane can be identified in rats under stereoscope and the neural injuries in different types of radical hysterectomy: sympathetic hypogastric nerve (HGN), parasympathetic pelvic splanchnic nerve (PSN), major pelvic ganglion (MPG), uterine nerve (UN), vesical nerve branch (VNB), and cavernous nerve (CN). A small rectal nerve branch can also be found in most rats, and vaginal/urethral nerves (VUN) can be identified in some rats. Abbreviation: UB: Urinary bladder. UR: Ureter. UT: Uterus. The red mark indicates resected neural tissue, and the dotted red line indicates crashed neural tissue during operation. **B**, pelvic view and specimens of type A, B1, C1, C2 and C2N groups. *: vagina, arrow: ureter, arrow head: paracervical tissue after resection, dashed line: the site of pathological examination.

vesicocervical ligament was resected. Then the uterine horn was separated from the ovary with the same procedure, and the mesometrium was resected at the level of ureter. The uterine artery and vein were cut where they crossing the ureter. The peritoneum of Douglas Pouch was opened and the rectum was separated from cervix and vaginal fornix. The lateral parametrium media to the ureter and an 1–2 mm long of vagina cuff were resected, and the vaginal cuff was closed with 6–0 prolene.

(3) For type C1 RH, a peritoneal incision was made media to the mesoureter, and the ureter was freed from parametrium. The bladder was separated from the cervix and vaginal fornix after opening vesicouterine peritoneal reflection, and the vesicovaginal space was created by blunt dissection with VNB preserved in the lateral fascia. Another peritoneal incision was

made lateral to the mesoureter, and the parametric adipose tissue was separated to expose the pelvic plexus. The deep uterine vein was resected after coagulation near the internal iliac vein. HGN, PSN and MPG were exposed and separated from the parametrium laterally, the uterine nerve (UN) was resected near MPG, and the pelvic plexus formed a PNP after the dissection. The vesicocervical ligament containing superior vesical vessels was separated and resected near the bladder, and the vaginal vein and vesicovaginal ligament were resected after coagulation approximately 2–3 mm caudal to the fornix. Then uterine horn was separated from the ovary with the same procedure, and the mesometrium was resected approximately 2 mm lateral to the ureter with HGN preserved. The uterine artery and vein were resected after coagulation near the internal iliac vessels. The peritoneum of Douglas Pouch was opened and the rectum was separated from cervix and vagina. The lateral parametrium was resected at the rectum but sparing PNP, and the vagina cuff was resected with a length of 2–3 mm, and vaginal cuff was closed with 6–0 prolene.

(4) For type C2 RH, the procedures including freeing ureter, separating bladder from cervix and vagina, separating parametric adipose tissue to expose the pelvic plexus, deep uterine vein resection, vesicocervical ligament resection, vaginal vein and vesicovaginal ligament resection, and separating uterine horn were all done by the same manipulation as described in type C1 RH. Then the mesometrium was resected near pelvis, and uterine artery and vein were resected after coagulation near the internal iliac vessels. The peritoneum of Douglas Pouch was opened and the rectum was separated from cervix and vagina. The lateral parametrium was resected at the sacrum and alongside the medial aspect of the internal iliac vessels, resulting in a complete removal of PNP composed by HGN, PSN, MPG, UN, VNB, and CN. The vagina was resected with a length of approximately 3 mm, and vaginal cuff was closed with 6–0 prolene.

(5) For type C2N, most procedures were manipulated similarly with those in type C1 RH, including freeing the ureter, separating the bladder from the cervix and the vagina, separating the parametric adipose tissue to expose the PNP, and resecting the deep uterine vein. Transection was made at HGN approximately 5 mm from the MPG, at PSN near the internal iliac vein, at UN between the MPG and the cervix, at VNB near the ureterovesical junction, at the CN approximately 2–3 mm from the MPG, thereafter, the en bloc excision of the PNP was successfully performed.

(6) For the sham group, only the same mid-line incision was made.

At the end, the pelvic cavity was flushed with normal saline and the abdominal incision was closed with 4–0 prolene. The operation time and blood loss were recorded.

The landmarks of ureter management, paracervical resection, ventral parametrial resection, dorsal parametrial resection, and vaginal resection were summarized in Table 1.

### 2.4 Residual urine volume measurement

Urine was extruded into a collecting cup by transabdominal manual compression and measured at random time on 3, 7, 14, and 21 days after operation. Transabdominal manual compression was performed twice a day to empty the bladder when rats had a residual urine volume > 5 ml.

### 2.5 Urine spotting test

The rats were placed individually in a 19×27 cm cage with a grid surface where they could drink tap water and food at will (Fig 3B1). Their urine spots were measured on paper filter for 6 h after a habituation period of 12 h. The paper was photographed under ultraviolet light and analyzed using Image J software.

**Table 1. The main landmarks in each type of radical hysterectomy in rats.**

| Type | Ureter | Resection of paracervix | Ventral parametrium | Dorsal parametrium | Vaginal resection |
|------|--------|-------------------------|---------------------|--------------------|-------------------|
| A | Directly visualized | Halfway between the cervix and ureter | Minimal excision | Minimal excision | Minimal excision |
| B1 | Unroofed and mobilized | At the ureteral tunnel | Partial excision of the vesicouterine ligament | Partial resection of the uterosacral peritoneal fold | 1–2 mm |
| C1 | Unroofed and mobilized until exposing HGN | At the iliac vessels (PSN and MPG are preserved) | Excision of the vesicouterine ligament at the bladder and proximal part of the vesicovaginal ligament (VNB is preserved) | At the rectum (HGN is preserved) | 2–3 mm |
| C2 | Unroofed and mobilized until exposing HGN | At the iliac vessels, (PSN and MPG are sacrificed) | At the bladder (VNB is sacrificed) | At the sacrum (HGN is sacrificed) | 3 mm |
| C2N | Unroofed and mobilized until exposing HGN | PNP and deep uterine vein are resected | Some fascia alongside VNB is resected | Not applicable | Not applicable |

Abbreviations: HGN: hypogastric nerve, MPG: major pelvic ganglion, PNP: pelvic neural plane, PSN: pelvic splanchnic nerve, VNB: vesical nerve branch.

## 2.6 Awake cystometry and leakage point pressure (LPP) measurement

Two fine silicon catheters, one for fluid infusion and the other one for pressure recording, were implanted at the bladder dome and tunneled to the interscapular region. After a 24-h recovery, the rats were placed in the metabolic cage (Fig 3B2). After 30-min habituation, intra-vesical pressure was recorded for a total of 40 min: 5 min before infusion, 30 min during infusion (infusion rate: 100 μl/min), and 5 min post infusion. Urine output along with the time was recorded, and the following cystometric parameters were analyzed: amplitude of contraction peak ($cmH_2O$), mean voided volume (g) and voided interval (second).

After cystometry, the rat was anesthetized with isoflurane and placed supine at zero pressure level and LPP was measured with half bladder capacity. The peak intravesical pressure when leakage occurred was recorded as LPP. LPP measurement was repeated 3 times and the average value was used.

## 2.7 Histological examination

Five μm-thick sections were prepared from paraffin-embedded neural and bladder tissues, routinely dewaxed in water, followed by H&E and Masson's trichome staining using the standard protocol. The areas of different vesical layers and collagen volume fraction (CVF = average collagen area/area of total field×100) were analyzed with Image J Software.

## 2.8 Immunohistochemistry

Paracervical neural sections were blocked with 3% hydrogen peroxide solution after antigen retrieval. Slides were incubated with anti-protein gene product 9.5 (Pgp9.5, a pan-neuronal marker [10, 11]) antibody (1:100, ab108986, Abcam, UK) for the detection of neural tissues for 45 min, and subsequently with a peroxidase-labelled second antibody for 30 min, 3,3-diaminobenzidine chromogenic solution for 10 min and hematoxylin for 30 sec.

## 2.9 Statistical analysis

All data were analyzed using SPSS 22.0 statistical software. Descriptive statistics were reported as mean ± standard deviation. One-way analysis of variance (ANOVA) was applied to compare the means of the samples between different groups. $0.01 \leq p < 0.05$ indicates a significant difference, marked by *, $0.001 \leq p < 0.01$ was marked by **, and $p < 0.001$ was marked by ***.

## 3. Results

### 3.1 Establishment of type A, B1, C1, C2 models

The operation time was 43.6±5.3 min, 54.9±5.1 min, 76.3±10.1 min, 58.4±2.2 min and 56.6 ±8.0 min for type A, B1, C1, C2, and C2N groups, respectively. Type C1 RH was the most complicated procedure and cost the longest operation time ($p<0.05$) compared with the other 4 groups, and there was no difference between type C2 and C2N groups. One rat in type B1 group, 2 rats in type C1 group and 7 rats in type C2 group died within 10 days after operation, thus type C2N was used as alternative to type C2 RH for further evaluation due to its similar operation time, neural excision and urinary retention but a favorable survival rate.

As shown in Fig 1B, the ureter was exposed in type B1, C1, C2 and C2N groups; The PNP was preserved but not visible in type A and B1 groups, preserved and visible in type C1, and excised in type C2 and C2N groups.

### 3.2 Paracervical excision size detected on H&E and Pgp9.5 staining

A gross examination of specimen showed gradual increase in resected paracervical tissue from type A to C2 groups (Fig 1B). As shown in Fig 2, H&E staining showed gradual increase in resected paracervical tissue from type A to C2 groups. a quantitative analysis based on Pgp9.5 staining demonstrated that the excised paracervical neural tissue area in C2 group was 0.0455 ±0.0204mm$^2$ on slide, which was significantly larger than that in type A (0.0024±0.0010mm$^2$), B1 (0.0054±0.0018mm$^2$), and C1 (0.0079±0.0057mm$^2$) groups. Compared with type A (0.0553 ±0.0162mm$^2$) and B1 (0.0917±0.0359mm$^2$) groups, the excised paracervical facial area was significant high in type C1 (0.1546±0.0481mm$^2$) and C2 (0.1539±0.0586 mm$^2$) groups. There was no difference in excised paracervical facia or total area between type C1 and C2 group.

### 3.3 Voiding pattern after RH

The residual urine volume was increased with the aggravation of neural injury, and was highest in type C2N at each time point (Fig 3). In type C1 group, it peaked at 7 days after operation, and then gradually decreased, which reflected some self-recovery. One rat in type B1 group (1/10), 4 rats in type C1 group (4/10) and all rats in type C2N group (10/10) experienced residual urine volume >2.0 ml, indicating impaired voiding regulation in these rats. As shown in Fig 3, a competent voiding in sham group resulted in large and aggregated spots. As the neural injury aggravated, the spots became smaller and more discrete from type A to C2N groups, which indicated impaired voluntary micturition and voiding habit.

### 3.4 Cystometry at 21 days after RH

The loss of neural innervation would impair the sensation, autonomic dilation and contraction of bladder, caused urinary retention and instable detrusor with overactive bladder, characterized by overflow incontinence and involuntary nonvoiding contractions (NVCs). While sham-operated controls displayed regular micturition cycles without NVCs, all rats in type C2N group (5/5) and C2 group (3/3) exhibited severe overflow incontinence and NVCs. NVCs with intermittent overflow incontinence were also observed in type B1 (1/5) and C1 groups (2/5). As shown in Fig 4, the amplitude of contraction peak was significantly reduced in type C2N group, revealing an underactive bladder. The mean voided volume was significantly increased in type C1 group, consistent with its high bladder compliance and longest voided interval. The LPP was significantly decreased in type B1, C1, and C2N groups, which indicated a loss of neural/structural support for continence. Cystometrograms also demonstrated detrusor instability

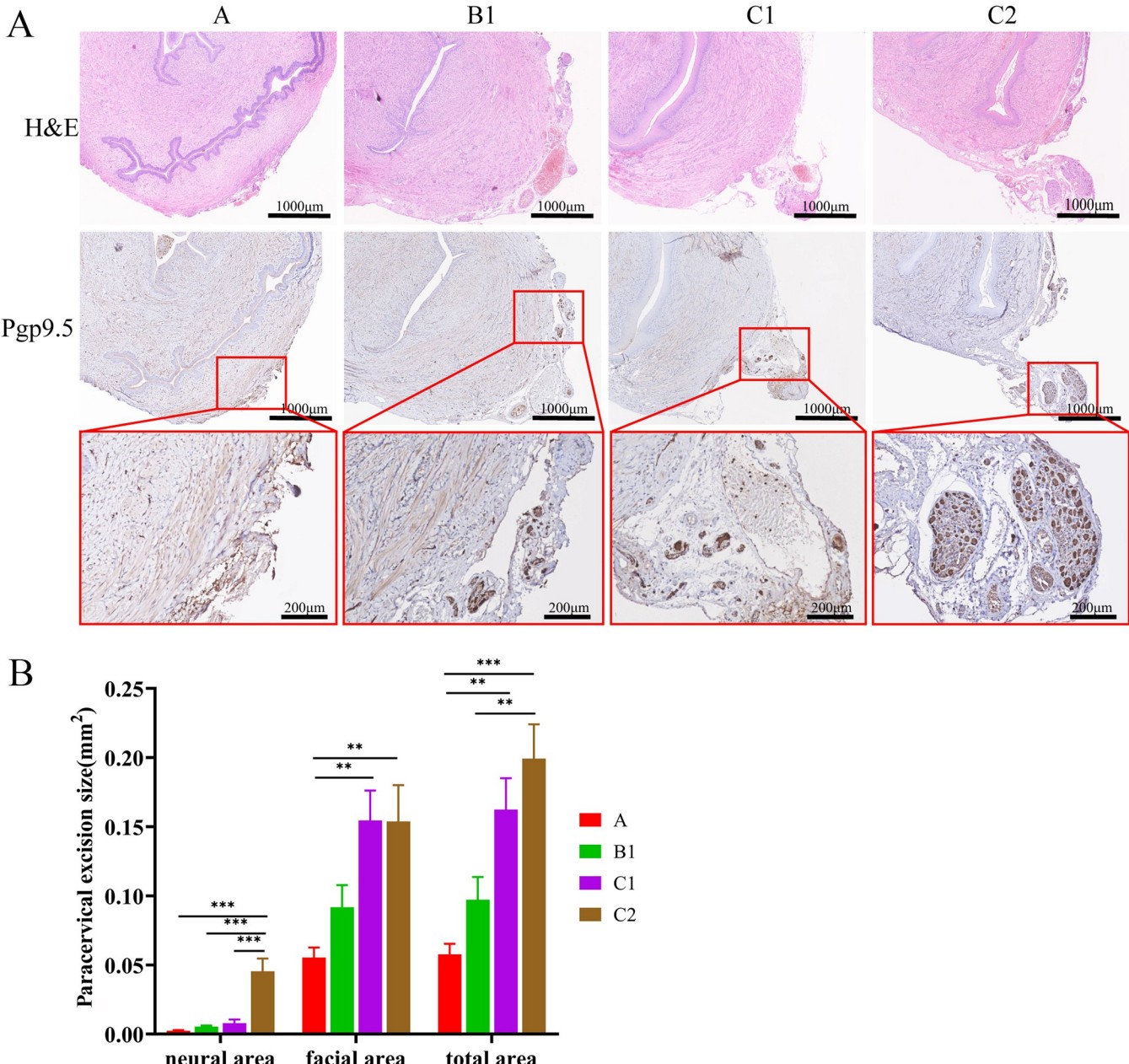

**Fig 2. Quantitative analysis of resected neural and facial tissue size of paracervix. A**, hematoxylin and eosin (H&E) and Pgp9.5 staining for neural tissues. **B**, the size of paracervical neural and facial excision. Reprinted from "Pelvic neural injuries and acute voiding changes in rat models of radical hysterectomy" under a CC BY license, with permission from Yanpeng Wang, original copyright 2024.

in some rats in type A, B1, and C1 groups. The incidence, frequency and severity of detrusor instability increased as the excision and injury extended (Fig 4E).

### 3.5 Bladder structural changes at 21 days after RH

As the surgical injuries aggravated, the bladder size, residual uterine volume and bladder weight were gradually increased (Fig 5). Bladder pathology showed minimal to mild compensatory changes in type A and B1 groups. Most of the rats in type C1 group showed mild to

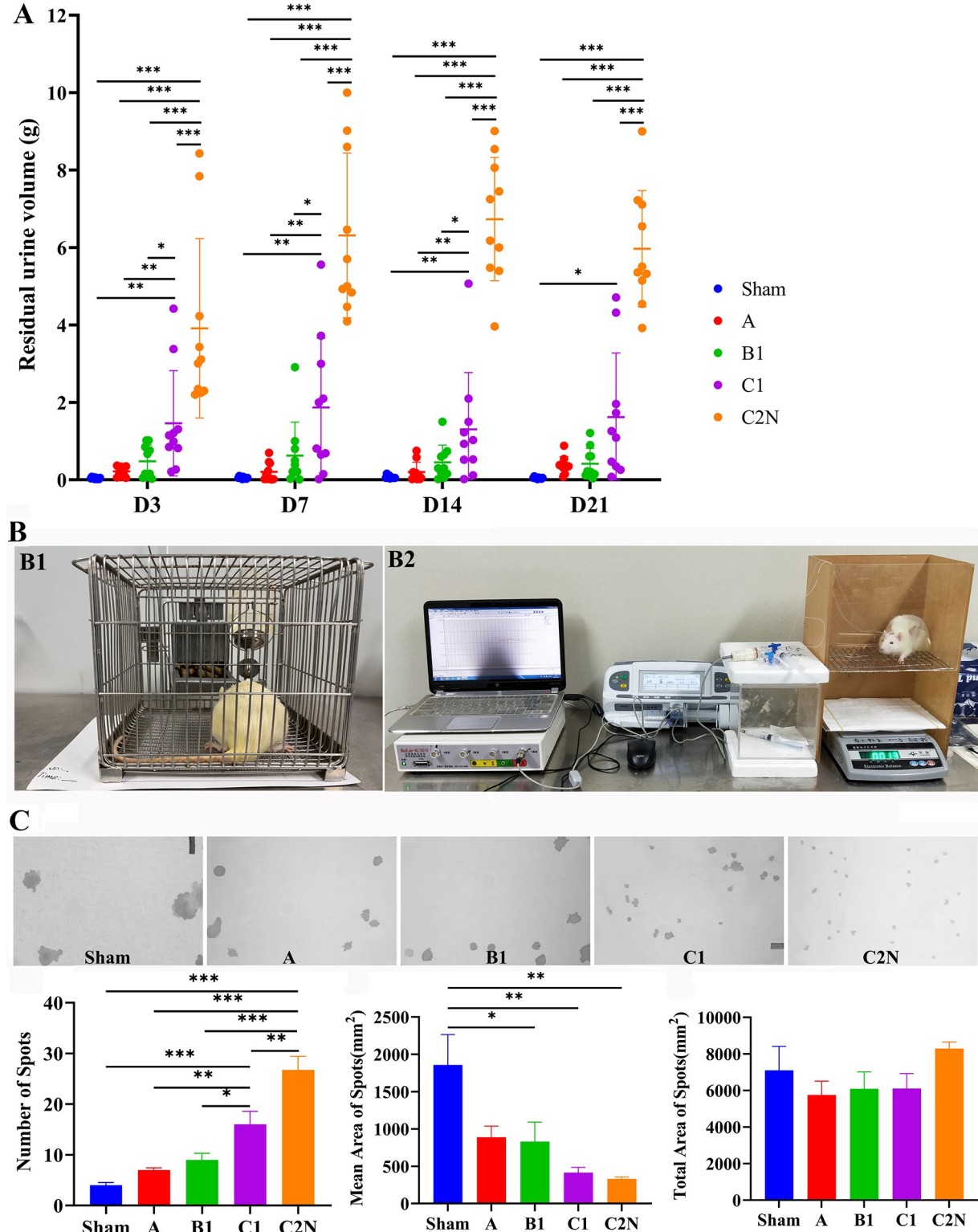

**Fig 3. Voiding behavior postoperatively. A**, residual urine volume. **B1**, the procedure of urine spotting test. **B2**, the procedure of cystometry. **C**, the results of urine spotting test. Reprinted from "Pelvic neural injuries and acute voiding changes in rat models of radical hysterectomy" under a CC BY license, with permission from Yanpeng Wang, original copyright 2024.

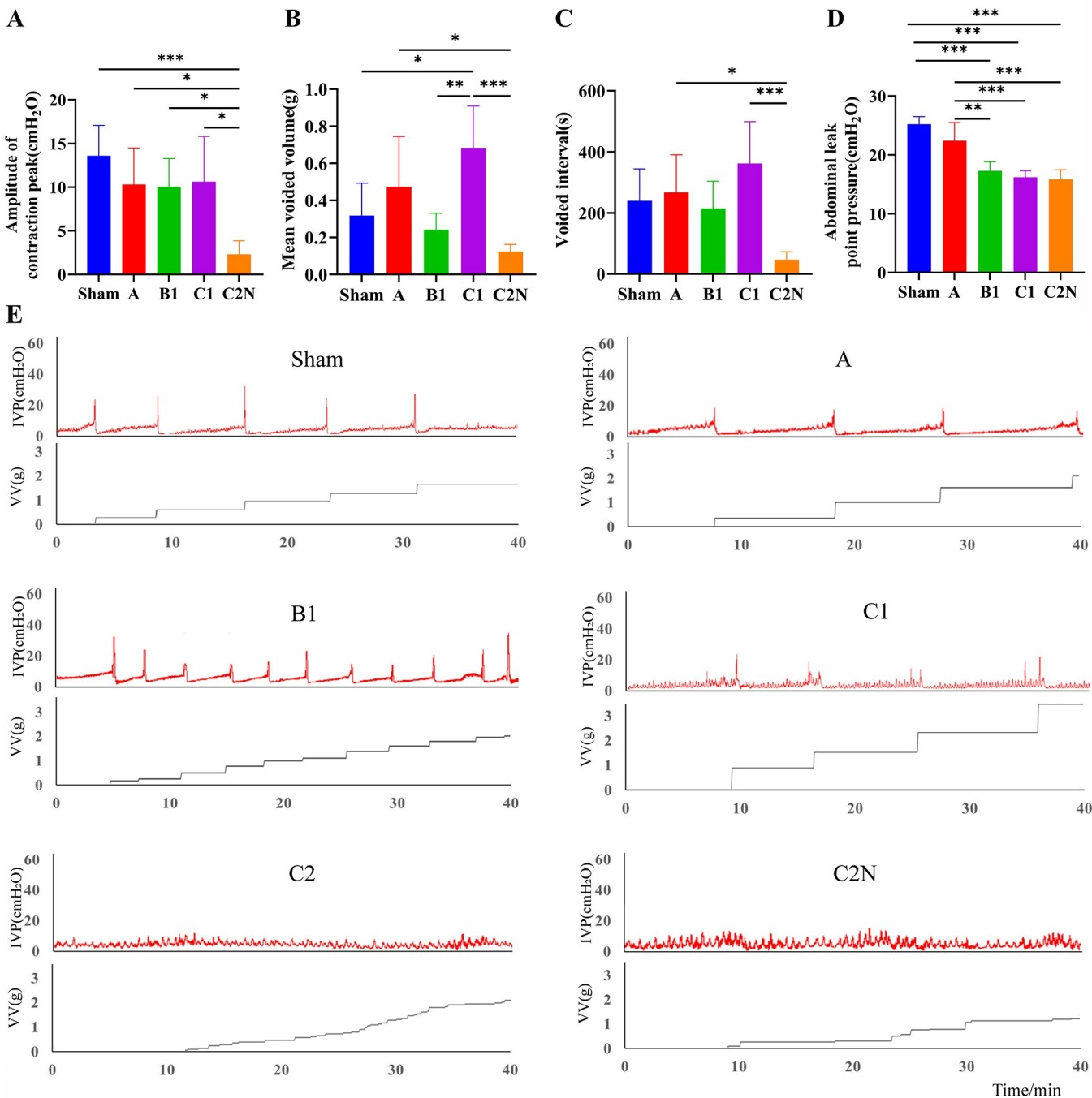

**Fig 4. Urodynamic characteristics measured by cystometry at 21 days postoperatively.** IVP: Intravesical pressure, VV: Voided volume.

moderate compensatory changes of bladder, while all rats in type C2N group revealed significant enlarged bladder, obviously thicken mucosa and adventitia, and stromal fibrosis. The collagen deposition of bladder, measured by CVF, was gradually increased as the surgical injuries aggravated. The average bladder weight in sham group was 0.14±0.02g, and there were 0/5, 1/5, 2/5 and 5/5 rats in type A, B1, C1 and C2N groups with bladder weighted two folds heavier, which indicated neurogenic bladder in these rats.

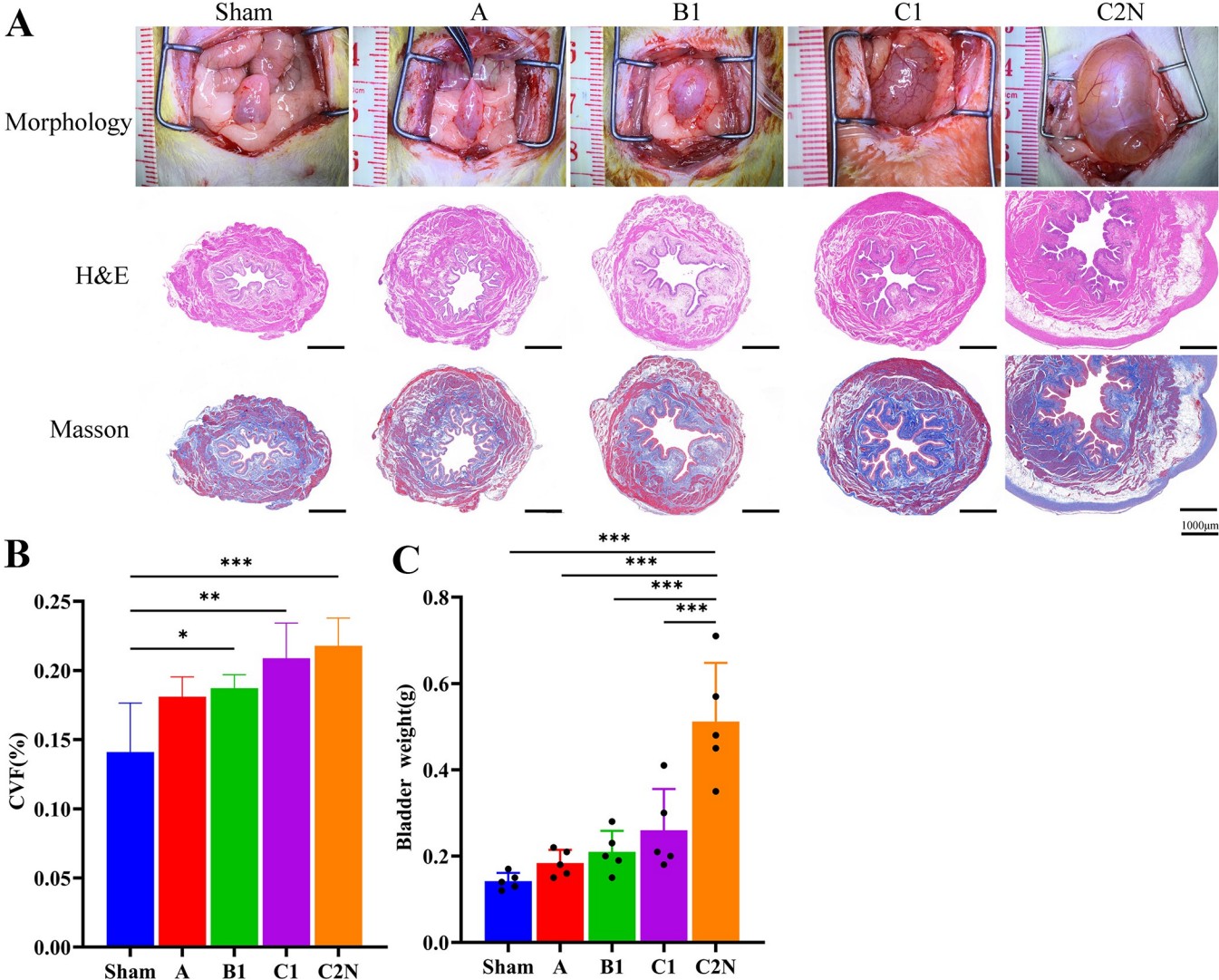

**Fig 5. Structural changes of bladder at 21 days postoperatively. A,** macroscopic view, hematoxylin and eosin (H&E) and Masson's staining of bladder. **B,** bladder weight, **C,** collagen volume fraction. Reprinted from "Pelvic neural injuries and acute voiding changes in rat models of radical hysterectomy" under a CC BY license, with permission from Yanpeng Wang, original copyright 2024.

### 3.6 PNP pathology at 21 days after RH

The PNP could be identified under stereoscope in sham, type A, B1 and C1 groups at 21 days after operation, while no obvious neural structure was found in type C2N group. As shown in Fig 6, the nerve fibers and neurons in sham group showed regular staining without degeneration on H&E staining. In type C2N group, the neural structure was removed and only facial tissue could be detected. In type A group, HGN, PSN and VNB showed minimal changes, and neuronal degeneration was occasionally detected in MPG. In type B1 group, some neural fibers showed edema with axonal degeneration, and several neurons in MPG displayed a degenerative phenotype. In type C1 group, the fibers and neurons demonstrated typical signs of fiber edema, axonal degeneration and neuronal degradation, which indicated a more intensive injury. The severity of neural damage was consistent with the extension of surgical excision and urinary pattern.

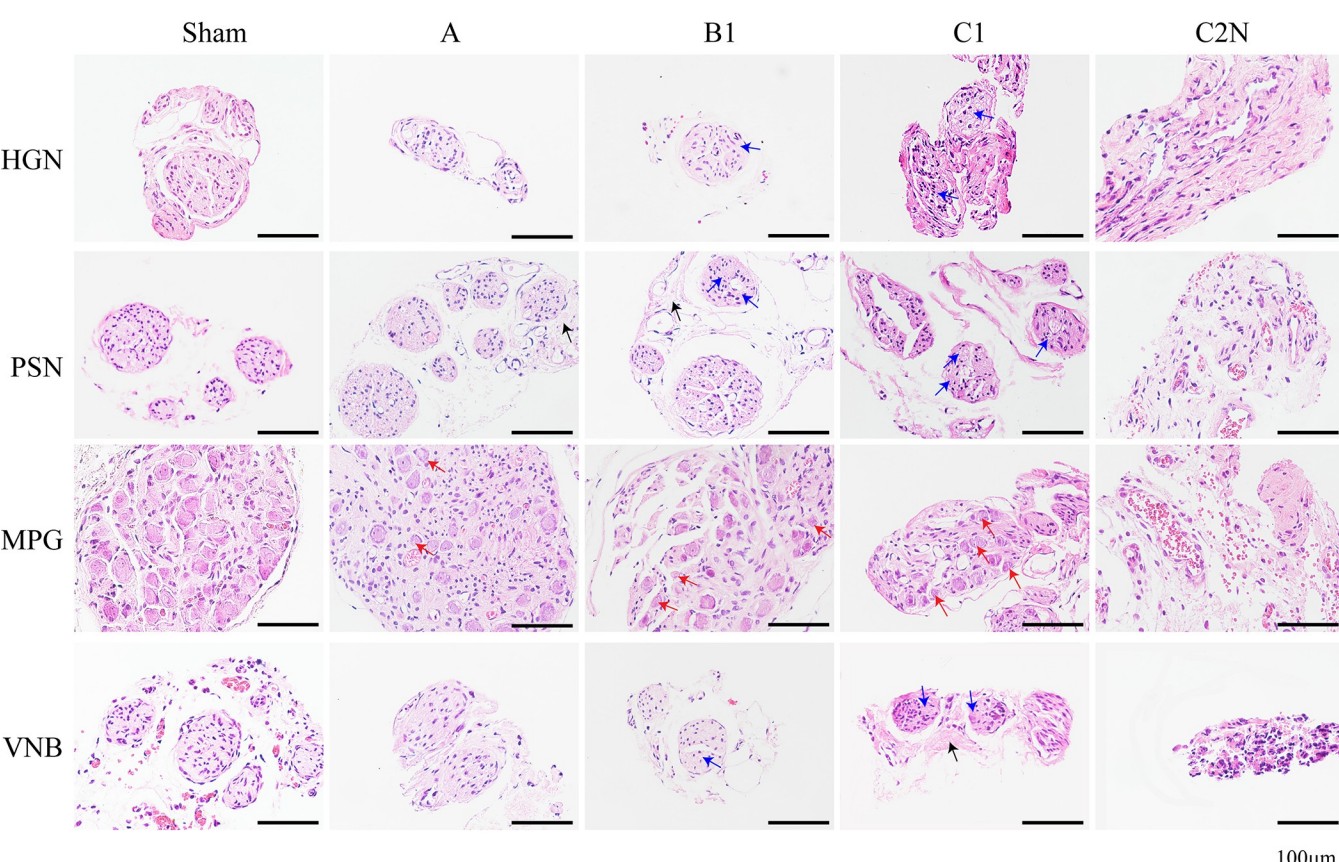

**Fig 6. H&E staining of HGN, PSN, VNB and MPG at 21 days postoperatively.** Red arrow: neuronal degeneration. Black arrow: nerve fiber edema. Blue arrow: axonal degeneration. Abbreviations: HGN: hypogastric nerve, MPG: major pelvic ganglion, PSN: pelvic splanchnic nerve, VNB: vesical nerve branch. Reprinted from "Pelvic neural injuries and acute voiding changes in rat models of radical hysterectomy" under a CC BY license, with permission from Yanpeng Wang, original copyright 2024.

## 4. Discussion

In this study we aimed to develop representative rat models of RH according to Q-M classification, and get insight into the neural and vesical changes of voiding after RH. We showed that type A and B1 RHs excised less PNP and resulted in minimally and mildly impaired voiding function. Type C2 RH excised PNP extensively and led to severe urine retention and voiding dysfunction. Moreover, a nerve-sparing type C1 RH preserved PNP and protected voiding function to some extent. A gradual increase in collagen deposition, bladder weight, together with decrease in area of urine spot and mean contraction peak pressure, nicely revealed the gradual structural and functional deterioration from types A to C2 RHs. Standard and reproducible animal models of RH were established in this research and could mimic the structural and functional changes after RHs in humans. These rat models could be applied for pelvic neurological studies and further development of drug/material treatment for neurogenic bladder dysfunction following RHs.

Lower urinary tract dysfunction is often overlooked in type A and B1 RHs because of the low incidence of urinary retention in these patients, and our findings revealed some clues of neurological damage and voiding dysfunction in type B1 RH, as manifested by decreased mean area of urine spots and leakage point pressure, and increased collagen volume fraction in affected rats. Besides, nerve-sparing type C1 RH preserved pelvic neural structures with comparable paracervical facial resection and improved bladder recovery compared with type C2 models. Indeed, NSRH is increasingly used in clinical practice, as the urodynamic

assessment in patients with NRSH has presented better results [12, 13]. Furthermore, some clinical studies with short or long-term follow-up have shown that NSRH appears to improve bladder function without compromising overall survival [14, 15]. We used a PNP separating technique to preserve the neural structures for voiding in rats, however, the neural injuries still caused up to 50% obviously urinary impairments (manifested by residual urinary volume >1.5 ml on day 7 postoperatively). Several nerve-sparing procedures have been reported in human, including electrical stimulation monitoring [16], partial preservation of the pelvic nerve plexus [17], neural dissection with waterjet [18], and robot-assisted laparoscopic RH [19].

Lower urinary tract dysfunction after RH is common but poorly understood [20]. The multifactorial etiology complicates the development of animal models to study this bladder disorder. Pelvic neural structures may be disrupted due to resection of uterosacral and rectovaginal ligaments, the dorsal and lateral paracervix, the caudal part of the vesicouterine ligaments, and the vagina [21, 22]. The neural injury could be a complete transection and resection of the nerves and ganglia with no potential to regenerate, and/or a transection of the axon with an intact nerve sheet allowing some regeneration, and/or a neurapraxia of temporary blockade of signal transmission without axon lesion [23]. Indeed, pelvic neural crush [24] or transection [25] could cause bladder dysfunction and overflow incontinence in rats. Therefore, the neural injuries could attribute to a combination of surgical excision, crush, stretch, electrothermal injury, ischemia, and inflammation. Our neuropathological findings showed that the main components of PNP involved in neural injury included PSN, HGN, MPG and VNB. Strategies targeting all components of pelvic neural plexus should be developed to reduce the neural injuries in RHs in the future.

We also found decreased LPP in rats post RH, which would raise more concerns about *de novo* urinary incontinence in addition to urinary retention after RH in humans. Urinary incontinence is a common complication after RH [26], as proved by significantly lower LPP in type B1, C1 and C2 groups in present study. In addition to partial vaginal resection, pelvic nerve injury has been shown to impair female genital blood flow and induce vaginal fibrosis [27], which may further weaken the posterior support to urinary continence.

This study was limited by the fact that although our preliminary experiment of type C1 RH revealed a similar change of voiding 3 months postoperatively compared to 21 days postoperatively, in this study we focused only on the early evaluation of voiding function within 21 days after operation. Besides, compared with humans, the lack of catheterization and inability to urinate with Valsalva maneuver in rats might impair the bladder structure and function, leading to worse urination outcomes. In addition, we have found that in rats, PSN aroused from L6 and S1 spinal nerves, and formed one bundle of nerves at paracervix and then inserted into MPG (unpublished data). However, in humans, PSNs arise from S2, S3, and S4 sacral nerves and run in three separated bundles at paracervix. Thus, type C2 RH in rats could remove PSN completely and results in severe urinary retention. In contrast, type C2 RH in humans might resect less PSNs, thus preserves some urinary structures and leads to favorable voiding function.

## Supporting information

**S1 Data.**
(XLSX)

## Acknowledgments

We gratefully acknowledge Dr Weijiao Fan from Zhejiang Provincial Hospital, for his help with animal experiment, and Drs. Wei Wang and Xin Zhang from Zhejiang Provincial Hospital, for their expertise and assistance with pathological examination.

## Author Contributions

**Conceptualization:** Haibin Wei, Jing Zhang, Yanpeng Wang.

**Data curation:** Miaomiao Liu, Lin Qian, Ting Wang.

**Formal analysis:** Miaomiao Liu, Ting Wang.

**Funding acquisition:** Yanpeng Wang.

**Investigation:** Miaomiao Liu, Lin Qian, Haibin Wei, Ting Wang.

**Methodology:** Miaomiao Liu, Lin Qian, Haibin Wei, Ting Wang.

**Project administration:** Ting Wang.

**Software:** Ting Wang.

**Supervision:** Yanpeng Wang.

**Validation:** Miaomiao Liu, Lin Qian, Ting Wang.

**Visualization:** Miaomiao Liu, Lin Qian, Ting Wang.

**Writing – original draft:** Miaomiao Liu, Lin Qian, Jing Zhang, Ting Wang, Yanpeng Wang.

**Writing – review & editing:** Jing Zhang, Ting Wang, Yanpeng Wang.

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
