## [Decision Letter · Decision Letter 0]

14 May 2024

PONE-D-23-42513Pelvic neural injuries and acute voiding changes in rat models of radical hysterectomyPLOS ONE

Dear Dr. Wang,

Thank you for submitting your manuscript to PLOS ONE. After careful consideration, we feel that it has merit but does not fully meet PLOS ONE’s publication criteria as it currently stands. Therefore, we invite you to submit a revised version of the manuscript that addresses the points raised during the review process.

The methodology and results of the manuscript require additional clarifications and explanations. The reviewers comments must be addressed suitably.

We look forward to receiving your revised manuscript.

Kind regards,

Antoine Naem, M.D.

Academic Editor

PLOS ONE

Journal Requirements:

2. To comply with PLOS ONE submissions requirements, in your Methods section, please provide additional information regarding the experiments involving animals and ensure you have included details on  methods of sacrifice and efforts to alleviate suffering.

 [National Natural Science Foundation of China (81971363), Natural Science Foundation of Zhejiang Province (LY22H040008), and Zhejiang Traditional Chinese Medicine Administration (2021ZB021).].  

5. We note that Figure(s) 1, 2a, 3b, 5a and 6 in your submission contain copyrighted images. All PLOS content is published under the Creative Commons Attribution License (CC BY 4.0), which means that the manuscript, images, and Supporting Information files will be freely available online, and any third party is permitted to access, download, copy, distribute, and use these materials in any way, even commercially, with proper attribution. For more information, see our copyright guidelines: http://journals.plos.org/plosone/s/licenses-and-copyright.

a. You may seek permission from the original copyright holder of Figure(s) 1, 2a, 3b, 5a and 6 to publish the content specifically under the CC BY 4.0 license. 

Reviewers' comments:

Reviewer's Responses to Questions

**Comments to the Author**

1. Is the manuscript technically sound, and do the data support the conclusions?

Reviewer #1: Yes

Reviewer #2: Partly

2. Has the statistical analysis been performed appropriately and rigorously? 

Reviewer #1: Yes

Reviewer #2: Yes

3. Have the authors made all data underlying the findings in their manuscript fully available?

Reviewer #1: Yes

Reviewer #2: Yes

4. Is the manuscript presented in an intelligible fashion and written in standard English?

Reviewer #1: Yes

Reviewer #2: Yes

5. Review Comments to the Author

Reviewer #1: It is an interesting work, the authors spend a remarkable effort to obtain their results, the result was well presented, some remarques required a major revision

In abstract

Add a brief explanation to compare the various terms of types A, B1, C1 and C2.

Add (using immunohistochemistry) in methodology

It is better to use passive tense rather using (We established …..)

In introduction

Line 49: Add a paragraph to compare the various terms of types A, B1, C1 and C2

Line 54: Add a paragraph to explain the normal anatomy of pelvic plexus in human and to justify the use of rat model with references

The paragraph between lines 59 and 61 need a suitable reference

Line 62: (Not only did we show the structural) needs lingual revision, native speaker revision is demanded.

In results

It is not clear whether the loose of rats in B1 and C1 was related to the operation complication? Please clarify.

Line 98: Adding schematic drawing to explain the different types of procedure with location of nerves will help to explain.

Line 171: Adding samples of images of 2:5 and 2.6 procedures could help to explain

Line 195: Add more information about the primary and secondary antibodies and the action of Pgp9.5. with reference>

In discussion:

The clinical application according to the results of this work was not obvious, is it the standardizing of the model or the application in surgical technique in human.

Reviewer #2: The author describes the establishment of a new animal model mimicking neurogenic bladder dysfunction induced by pelvic nerve injury following radical hysterectomy. The new rat model was characterized by the severity of bladder dysfunction being graded by surgical radicality, which is interesting. However, this manuscript still has some points to be resolved, as follows:

Major

1) Several surgical procedures have been developed to create a stratified neurogenic bladder in rats mimicking UAB followed by total hysterectomy. How should the author decide what millimeters of tissue from cervices should be dissected in each rat model?

2) The author mentioned that residual urine was measured by pressing the lower abdomen (P.9, line 167). Is this reliably reproducible?

3) All C2N and C2 rats exhibited NVCs in the CMGs despite being subjected to severe pelvic nerve injury by wider resection. Why? Discuss the issue (P.12, line 240).

Minor

1) how do you evaluate QOL of rat? ( P.12, line 237)

6. PLOS authors have the option to publish the peer review history of their article (what does this mean?). If published, this will include your full peer review and any attached files.

Reviewer #1: No

Reviewer #2: No

---

## [Author Response · Author response to Decision Letter 0]

28 Jul 2024

Reviewer #1: It is an interesting work, the authors spend a remarkable effort to obtain their results, the result was well presented, some remarques required a major revision

In abstract

Add a brief explanation to compare the various terms of types A, B1, C1 and C2. 

Add (using immunohistochemistry) in methodology.

It is better to use passive tense rather using (We established …..)

In introduction

Line 49: Add a paragraph to compare the various terms of types A, B1, C1 and C2. 

Line 54: Add a paragraph to explain the normal anatomy of pelvic plexus in human and to justify the use of rat model with references.

The paragraph between lines 59 and 61 need a suitable reference.

Line 62: (Not only did we show the structural) needs lingual revision, native speaker revision is demanded.

Response: Thank you for your valuable advice. We have revised the manuscript according to the reviewer’s suggestion.

In results

It is not clear whether the loose of rats in B1 and C1 was related to the operation complication? Please clarify. 

Response: Thank you for your valuable advice. We have added this information in the section of “Experimental design”: Besides, one rat in type B1 group and two rats in type C1 group died 2 to 3 days after operation due to operation complication, and additional rats were supplemented.

Line 98: Adding schematic drawing to explain the different types of procedure with location of nerves will help to explain.

Line 171: Adding samples of images of 2.5 and 2.6 procedures could help to explain.

Line 195: Add more information about the primary and secondary antibodies and the action of Pgp9.5. with reference> 

Response: The reviewer’s comment is well appreciated. We have revised the manuscript, figure 1 and figure 3 according to the advices.

In discussion:

The clinical application according to the results of this work was not obvious, is it the standardizing of the model or the application in surgical technique in human.

Response: The reviewer’s comment is well appreciated. We have added some comments on this topic at the end of the first paragraph in the discussion section.

Reviewer #2: The author describes the establishment of a new animal model mimicking neurogenic bladder dysfunction induced by pelvic nerve injury following radical hysterectomy. The new rat model was characterized by the severity of bladder dysfunction being graded by surgical radicality, which is interesting. However, this manuscript still has some points to be resolved, as follows:

Major

1) Several surgical procedures have been developed to create a stratified neurogenic bladder in rats mimicking UAB followed by total hysterectomy. How should the author decide what millimeters of tissue from cervices should be dissected in each rat model?

Response: The reviewer’s comment is well appreciated. The average vaginal length of women is approximately 10 cm, and the length of vaginal resection of 0~10mm, at least 10mm, and 15-20mm in human RH of types A, B and C RH respectively1. The average vaginal length of adult SD rats is approximately 15 mm; accordingly, the length of vaginal resection we used is minimal, 1-2mm, 2-3mm and 3mm for types A, B, C1 and C2 RH in this study.

2) The author mentioned that residual urine was measured by pressing the lower abdomen (P.9, line 167). Is this reliably reproducible?

Response: The reviewer’s comment is well appreciated. A more accurate method of measuring residual urine is bladder intubation after anesthesia, which is rarely used. Abdominal compression is a simple, noninvasive method that is comparable to ultrasound measurements in accuracy2, although it may result in a small amount of urine remaining in the bladder. Our data from the same animal at different time points also showed the stability of the measured values.

3) All C2N and C2 rats exhibited NVCs in the CMGs despite being subjected to severe pelvic nerve injury by wider resection. Why? Discuss the issue (P.12, line 240).

Response: The reviewer’s comment is well appreciated. In fact, frequent nonvoiding contraction is not a real contraction but the characteristic of instable detrusor after severe urinary retention and neurogenic bladder. We have added this context in the manuscript: “The loss of neural innervation would impair the sensation, autonomic dilation and contraction of bladder, caused urinary retention and instable detrusor with overactive bladder, characterized by overflow incontinence and NVCs.”

Minor

1) how do you evaluate QOL of rat? ( P.12, line 237)

Response: The reviewer’s comment is well appreciated. The “Quality of life” referred to the voiding habit and pattern reflected by spotting test. Thus, we changed “quality of life” to “voiding habit” for accuracy.

References

1. Querleu D, Morrow CP. Classification of radical hysterectomy. Lancet Oncol. Mar 2008;9(3):297-303. doi:10.1016/S1470-2045(08)70074-3

2. Sartori AM, Schwab ME, Kessler TM. Ultrasound: A Valuable Translational Tool to Measure Postvoid Residual in Awake Rats? Eur Urol Focus. Sep 15 2020;6(5):916-921. doi:10.1016/j.euf.2019.11.016

---

## [Decision Letter · Decision Letter 1]

16 Sep 2024

PONE-D-23-42513R1Pelvic neural injuries and acute voiding changes in rat models of radical hysterectomyPLOS ONE

Dear Dr. Wang,

Thank you for submitting your manuscript to PLOS ONE. After careful consideration, we feel that it has merit but does not fully meet PLOS ONE’s publication criteria as it currently stands. Therefore, we invite you to submit a revised version of the manuscript that addresses the points raised during the review process.

We look forward to receiving your revised manuscript.

Kind regards,

Antoine Naem, M.D.

Academic Editor

PLOS ONE

**Journal Requirements:**

**Additional Editor Comments:**

Please address the reviewer’s comment regarding the voiding dysfunction in the Type A group.

Reviewers' comments:

Reviewer's Responses to Questions

**Comments to the Author**

1. If the authors have adequately addressed your comments raised in a previous round of review and you feel that this manuscript is now acceptable for publication, you may indicate that here to bypass the “Comments to the Author” section, enter your conflict of interest statement in the “Confidential to Editor” section, and submit your "Accept" recommendation.

Reviewer #1: All comments have been addressed

Reviewer #2: All comments have been addressed

2. Is the manuscript technically sound, and do the data support the conclusions?

Reviewer #1: Yes

Reviewer #2: Yes

3. Has the statistical analysis been performed appropriately and rigorously? 

Reviewer #1: Yes

Reviewer #2: Yes

4. Have the authors made all data underlying the findings in their manuscript fully available?

Reviewer #1: Yes

Reviewer #2: Yes

5. Is the manuscript presented in an intelligible fashion and written in standard English?

**Reviewer #1:** Yes

**Reviewer #2:** Yes

6. Review Comments to the Author

**Reviewer #1:** the authors responded to reviewers comments and preformed the demanded modifications in the manuscript

**Reviewer #2: **The manuscript has now been revised accordingly. However, the following issues remain to be discussed:

In the Discussion section, the authors described “ our findings revealed some clues of neurological damage and voiding dysfunction in type A and B1 RHs,,, “ (line 310). In contrast, there were no significant differences in voiding functional parameters between the sham and type A groups (Fig.3 and 4). The authors cannot conclude that even a type A RH procedure causes voiding dysfunction. Please answer.

7. PLOS authors have the option to publish the peer review history of their article (what does this mean?). If published, this will include your full peer review and any attached files.

Reviewer #1: No

Reviewer #2: No

---

## [Author Response · Author response to Decision Letter 1]

17 Sep 2024

The reviewer’s comment is well appreciated. We have revised the manuscript as “our findings revealed some clues of neurological damage and voiding dysfunction in type B1 RH, as manifested by decreased mean area of urine spots and leakage point pressure, and increased collagen volume fraction in affected rats”.

---

## [Editor Report · Decision Letter 2]

23 Sep 2024

Pelvic neural injuries and acute voiding changes in rat models of radical hysterectomy

PONE-D-23-42513R2

Dear Dr. Wang,

We’re pleased to inform you that your manuscript has been judged scientifically suitable for publication and will be formally accepted for publication once it meets all outstanding technical requirements.

Kind regards,

Antoine Naem, M.D.

Academic Editor

PLOS ONE
---

## [Editor Report · Acceptance letter]

4 Oct 2024

PONE-D-23-42513R2 

PLOS ONE

Dear Dr. Wang, 

I'm pleased to inform you that your manuscript has been deemed suitable for publication in PLOS ONE. Congratulations! Your manuscript is now being handed over to our production team.

Kind regards, 

on behalf of

Dr. Antoine Naem 

Academic Editor

PLOS ONE